# Preanesthetic Administration of Trazodone Does Not Impact Anesthetic Recovery Scores in Horses

**DOI:** 10.3390/ani15192907

**Published:** 2025-10-06

**Authors:** Emmanuel Jerome Joseph, Lydia Love, Michelle Mayakis, Kelley Varner

**Affiliations:** 1Department of Molecular Biomedical Sciences, College of Veterinary Medicine, North Carolina State University, Raleigh, NC 27606, USAkmvarner@ncsu.edu (K.V.); 2College of Veterinary Medicine, North Carolina State University, Raleigh, NC 27606, USA

**Keywords:** anesthesia, horse, recovery, trazodone

## Abstract

**Simple Summary:**

Trazodone is commonly administered to hospitalized equine patients to manage unwanted behaviors. Trazodone has been reported to cause decreased muscle coordination, which could impact a horse’s ability to stand after an anesthetic event, with the anesthetic recovery period posing a high risk to equine patients. Records of horses undergoing anesthesia were evaluated to determine if trazodone administration prior to anesthesia impacted the recovery process. The results of this retrospective study demonstrate that trazodone did not negatively impact the recovery period, and as such, may be administered prior to anesthesia.

**Abstract:**

Trazodone is administered to hospitalized equine patients to aid in behavioral management, but the effects on the anesthetic recovery period have not been investigated. This study sought to determine if there is an association between trazodone administration and recovery quality, recovery time, need for sedation, or need for reversal agent administration. We hypothesized that there would be no difference in recovery scores, recovery time, additional sedation, or reversal agent administration between horses that received preanesthetic trazodone and horses that did not. Records were reviewed to identify horses undergoing orthopedic MRI between January 2022 and January 2025. Of these horses, 19 were administered trazodone prior to anesthesia and 38 horses that did not receive trazodone were selected as case-matched controls. Signalment, anesthetic drug protocol, complications during anesthesia, duration of anesthesia, duration of recovery, recovery scores, recovery complications, sedation agents, and reversal agents administered during recovery were recorded. Trazodone administration was not associated with a significant difference in recovery scores between groups. Trazodone administration was not associated with a difference in recovery time or differences in sedation and reversal agent administration. Trazodone administration prior to anesthesia in horses undergoing orthopedic MRI did not impact the recovery period.

## 1. Introduction

General anesthesia in horses is associated with higher morbidity and mortality than in other species, with the recovery period being a particularly challenging time. Perioperative mortality rates of 0.9% have been reported in horses undergoing non-colic procedures, with cardiopulmonary arrest, post-operative fracture, and myopathy commonly implicated as causes of death [1]. Equine perioperative mortality rates have decreased over time, but equine anesthetic events still carry more risk than in other species [2,3]. The recovery period is of particular concern, with the majority of equine anesthesia-related mortality occurring during this period [4,5,6]. Therefore, identifying factors that can impact recovery is essential to reducing negative outcomes associated with equine anesthetic events. Trazodone is an anxiolytic medication, used routinely in canine and feline patients for behavior modification [7,8]. Similarly, trazodone is administered to equine patients to minimize anxiety and modify unwanted behaviors associated with hospitalization and stall rest, but it can result in mild ataxia and decreased muscle coordination [9]. These effects could be problematic in the recovery period. Previous studies have explored the cardiovascular effects of trazodone administration prior to anesthesia [10], but there are currently no reports on the impact of trazodone administration on equine recovery.

The objective of this study was to determine if the administration of trazodone prior to anesthesia of healthy equine patients undergoing orthopedic magnetic resonance imaging (MRI) would impact recovery scores. Secondary aims included whether there would be differences in the duration of recovery, the use of additional sedatives, or the use of reversal agents in the recovery period in horses that received trazodone compared to horses that did not. We hypothesized that there would be no difference in recovery scores, recovery time, additional sedation, or reversal agent administration between horses that received trazodone preoperatively and horses that did not.

## 2. Materials and Methods

### 2.1. Case Selection

A retrospective cohort study design was used. Owner authorization for anesthesia and imaging procedures was obtained for all cases, but consent was not obtained for this retrospective study. Medical records of a single institution (North Carolina State University College of Veterinary Medicine) from January 2022 to January 2025 were reviewed to identify horses that were anesthetized for MRI of an orthopedic complaint (Figure 1). Horses were excluded if they underwent additional procedures after MRI, were euthanized prior to recovery, had an MRI that was not orthopedic in nature, or had incomplete medical records.

### 2.2. Anesthetic Management

Anesthesia, including recovery, of all horses undergoing anesthesia for MRI was managed primarily by a dedicated anesthesia technician or anesthesia resident under the direct supervision of a board-certified anesthesiologist of the American College of Veterinary Anesthesia and Analgesia. Anesthetic management, including the administration of sedatives or reversal agents in recovery, was approved by the overseeing anesthesiologist. Rope-assisted recoveries were completed for all horses.

### 2.3. Data Collection

Medical records were reviewed by two authors (EJ, MM), and data collected included patient demographics (age, breed, weight, sex), procedural information, anesthetic drug protocols, and anesthesia time. For cases that received trazodone, the dose, frequency, and timing of the last dose prior to anesthesia were recorded. Recovery time, recovery scores, and complications during recovery were recorded. Recovery scores utilized were standard for the facility (Figure 2). Sedative and reversal agent administration during the recovery period was also recorded. Recovery complications were defined as the occurrence of fractures, myopathy, neuropathy, airway obstruction, pulmonary edema, or failure to stand.

The presence of anesthesia-related complications was recorded. Hypoxemia was defined as PaO2 < 80 mmHg; hypotension was defined as MAP < 70 mmHg; hypertension was defined as SAP > 160 mmHg; bradycardia was defined as HR < 25 beats per minute (bpm); and tachycardia was defined as HR > 45 bpm. Other complications appreciated under anesthesia (e.g., electrolyte abnormalities, arrhythmias other than sinus bradycardia or tachycardia) were also recorded. Treatments for documented complications were recorded.

### 2.4. Data Analysis

An a priori power analysis using a mean recovery score difference of 1 point, with an alpha of 0.05 and a power of 80% suggested that 12 horses in the trazodone group and 24 horses in the control group would be sufficient to determine statistical significance. Statistical analysis was performed using R v. 4.5.0. Breed and sex were compared between both groups using Fisher’s exact test. Age, weight, and anesthesia time were compared between groups using an independent *t*-test. A binary logistic regression was performed to determine if trazodone administration was associated with the occurrence of hypotension. A binary logistic regression model was used to compare recovery scores between groups. An independent *t*-test was used to compare median recovery times. A multiple linear regression was conducted to examine the effect of trazodone administration, anesthesia duration, and presence of hypotension on recovery time. A Poisson regression was used to compare the number of sedatives needed during the recovery period between groups. A binary logistic regression was used to compare reversal administration during recovery. The significance level was set to 5% (*p* < 0.05).

## 3. Results

### 3.1. Cases and Controls

Record review identified 219 horses that had an anesthetic event for orthopedic MRI during the stated time period; of these horses, 19 were administered trazodone prior to anesthesia. The mean dose of trazodone administered was 3.3 mg/kg (1.9–5.1 mg/kg). Two control horses of a similar signalment undergoing MRI were selected for each case as a comparison group (38 horses) (Table 1).

### 3.2. Anesthetic Management

Anesthetic protocols varied by horse, and choices were made at the discretion of the anesthesiologist. Pre-medication protocols included xylazine (trazodone group: 18/19 horses 1 mg/kg [0.79–1.2 mg/kg]; control group: 38/38 control horses, 1 mg/kg [0.75–1.6 mg/kg]), acepromazine (trazodone group: 14/19 horses 0.016 mg/kg [0.01–0.02 mg/kg]; 20/38 control horses 0.016 mg/kg [0.01–0.024 mg/kg]), dexmedetomidine (trazodone group: 1/19 horses, 0.004 mg/kg), and butorphanol (trazodone group: 19/19 trazodone horses 0.02 mg/kg [0.016–0.021 mg/kg]; control group: 35/38 horses, 0.02 mg/kg [0.016–0.022 mg/kg]). All horses were induced with ketamine (trazodone group: 3.1 mg/kg [2.9–4 mg/kg]; control group: 3 mg/kg [2.14–3.7 mg/kg]) and propofol (trazodone group 0.5 mg/kg [0.46–0.51 mg/kg]; control group 0.5 mg/kg [0.3–0.53 mg/kg]). All horses were maintained on an infusion of ketamine, xylazine, and guaifenesin that was adjusted at the discretion of the anesthetist for transport into the MRI room. During this period horses received ketamine (trazodone group: 0.7 mg/kg [0.2–0.9 mg/kg]; control group: 0.6 mg/kg [0.18–1.6 mg/kg]) xylazine (trazodone group: 0.31 mg/kg [0.07–0.8 mg/kg]; control group: 0.32 mg/kg [0.09–0.8 mg/kg]), and guaifenesin (trazodone group: 31 mg/kg [14–79 mg/kg]; control group: 32 mg/kg [9–81 mg/kg]). Horses were then transitioned to maintenance with isoflurane and a dexmedetomidine infusion. The dexmedetomidine infusion rate (0.5–2 mcg/kg/min) and inhalant concentration (0.8–2%) were adjusted at the discretion of the primary anesthetist. Mean anesthesia time for trazodone and control groups were 141.9 ± 41.6 min (80–196 min) and 137.2 ± 25.3 min (96–287 min), respectively. Anesthesia time was not significantly different between groups (*p* = 0.753, 95% CI [0.874, 1.103]).

Complications that were experienced during general anesthesia included hypotension noted in both control and trazodone groups (16/38, 8/19, respectively). Hypoxemia (1/38) was noted in the control group. Sinus tachycardia (1/19), bradycardia (1/19), and hypocalcemia (1/19) were noted in the trazodone group. No horses in the control group experienced tachycardia, bradycardia, or hypocalcemia. Hypotension duration ranged from 5 to 10 min, and was treated and resolved with titration of dobutamine in all horses. The single horse in the control group with evidence of hypoxemia on arterial blood gas was treated with an alteration in ventilator settings and the hypoxemia was resolved prior to recovery. Hypocalcemia is resolved with supplementation of calcium gluconate. Bradycardia and tachycardia resolved without intervention.

### 3.3. Recovery

Horses administered trazodone had a mean recovery score of 4.7 ± 0.2. Horses in the control group had a mean recovery score of 4.8 ± 0.1. Comparison of recovery scores did not reveal a significant difference between horses that received trazodone compared to horses that did not (*p* = 0.226, OR = 0.424, 95% CI [0.102, 1.743]) (Table 2). None of the horses included in this study experienced any recovery complications.

Mean recovery time for horses administered trazodone was 62.8 ± 21.3 min (40–135 min). Mean recovery time for control horses was 59.3 ± 12.0 min (40–91 min). There was no significant difference in recovery times between groups (*p* = 0.601, 95% CI [0.907, 1.182]). There was no significant effect of trazodone administration (*p* = 0.583) or hypotension (*p* = 0.923) on recovery time. There was a significant effect of anesthesia time on recovery time (*p* = 0.008).

All horses in both groups received a sedative during the recovery time period. Thei initial sedative agent used was either dexmedetomidine (n = 51) or xylazine (n = 6). In 5/19 of horses receiving trazodone only a single sedative agent was administered during recovery (dexmedetomidine in 4/19 [1 mcg/kg, 0.8–1 mcg/kg], and xylazine in 1/19 [0.25 mg/kg]), compared to 17/38 of horses in the control group (dexmedetomidine in 14/38 [1 mcg/kg, 0.8–1 mcg/kg)] xylazine in 3/38 [0.36 mg/kg, 0.25–0.49 mg/kg]). A second sedative agent was administered to 13/19 horses that received trazodone, compared to 17/38 horses in the control group. Additional sedative agents included xylazine (13/19 horses administered trazodone [0.35 mg/kg, 0.16–0.6 mg/kg], 13/38 control horses [0.26 mg/kg, 0.1–0.5 mg/kg]), detomidine (single control horse, 0.003 mg/kg), acepromazine (one control horse, 0.018 mg/kg), and ketamine (one control horse, 0.16 mg/kg). 4/38 horses in the control group received a third sedative agent, compared to 1/19 in the trazodone group. All horses that were administered a third sedative agent were given acepromazine (trazodone group: 0.0015 mg/kg; control group: 0.015 mg/kg [0.009–0.024 mg/kg]). There was no significant difference (*p* = 0.577, 95% CI [0.618, 2.250]) in the administration of additional sedation between the two groups.

Reversal agents were administered to 5/19 of the horses in the trazodone group and 3/38 of the horses in the control group. All horses were administered atipamezole as the sole reversal agent. Mean dose of atipamezole administered to control horses was 0.045 mg/kg (0.02–0.06 mg/kg). Mean dose of atipamezole administered to the trazodone group was 0.037 mg/kg (0.005–0.07 mg/kg). There was no significant difference for reversal administration between trazodone and control groups (*p* = 0.073, OR = 4.167, 95% CI [0.902, 22.620]). While no significant difference in reversal administration between groups was identified, a trend indicating increased reversal agent administration during recovery of horses administered trazodone was noted.

## 4. Discussion

The results of this retrospective study show that trazodone administration prior to anesthesia for orthopedic MRI may not result in a significant difference in recovery scores, recovery time, need for additional sedation, or reversal agent administration between horses that received trazodone preoperatively and horses that did not, supporting our hypotheses. As previously stated, trazodone use in horses can lead to mild ataxia and decreased muscle coordination [9], creating concern for the risk of anesthesia recovery complications. The results of this study suggest that the utilization of trazodone in a hospital setting may not negatively affect the recovery period. Thus, trazodone administration prior to anesthesia may not increase patient morbidity in the recovery period.

All horses in this study received at least one sedative agent during recovery, with no difference between groups in the utilization of sedatives. In all horses, an alpha-2 adrenergic agonist was administered. Administration of alpha-2 agonists in the recovery period has been associated with improved recovery outcomes [11]. A majority of the horses that were administered trazodone were given an additional sedative, but fewer horses in the trazodone group were administered a third round of sedation in comparison to the control group. A single horse was administered ketamine in the recovery period; we suspect that this was due to movement during positioning for recovery. Horses administered trazodone in this study may have exhibited more unwanted behaviors during hospitalization than the control population, resulting in the administration of trazodone. Horse temperament has been implicated in impacting recovery scores, with more difficult temperaments correlating with poorer recovery scores [12]. Horses with more difficult temperaments may require additional sedative agents in the recovery period, and trazodone administration may have helped minimize this requirement. The lack of difference in the requirement for a sedative agent may provide additional support for trazodone administration prior to anesthesia, as administration of trazodone prior to anesthesia may limit the need for repeated administration of a sedative agent in the recovery period, since its sedative effects may still be present. Sedation after orally administered trazodone was noted to have a duration of 0.5 to 8 h in most horses [13]. Therefore, the sedative effects of trazodone may carry on into the post-anesthetic period for horses administered trazodone prior to anesthesia. In contrast to this, a previous study evaluating the effects of trazodone on sedation with xylazine revealed that trazodone administration did not reduce xylazine doses [14]. Further prospective research is needed to determine if trazodone has potential effects on sedative requirements in recovery.

There was no significant difference between the administration of reversal agents between groups in this study, although a potential difference may have been masked by a small sample size. Indeed, if one more horse in the treatment group had been administered a reversal agent, a significant difference between groups would have been identified. As such, administration of reversal agents to horses that have been administered trazodone may be warranted in the recovery period. All horses that received a reversal agent were administered atipamezole, which has been shown to be efficacious in reducing sedation caused by alpha-2 agonists [15]. Reversal agents were administered in these horses due to slow recovery or movement attempts that were not perceived to be as robust as anticipated. Trazodone has been shown to have a half-life in horses of 4–6 h after the first administration and 7–9 h after the 4th administration [16]. As sedation has been noted 4 h post-trazodone administration [9,17], the sedative effects of trazodone may still have been present during recovery. The sedative effects of trazodone in conjunction with the utilization of additional sedative agents, such as alpha-2 agonists, in the recovery period may have led to increased reversal agent use. Further evaluation of the use of sedative agents and reversal agents in the recovery period is required to better determine if trazodone has an effect on these factors.

There was no significant difference in recovery times between the control and trazodone groups. In this population, pre-anesthetic administration of trazodone did not appear to impact recovery time, as there was no significant difference in recovery time between groups. Additionally, trazodone administration was not found to have a significant effect on recovery time. Previous studies have shown that there is a correlation between recovery duration and recovery quality [18]. Horses that did not attempt to rise promptly after an anesthetic event had better recovery qualities. A longer recovery period may allow a horse to reduce inhalant anesthetic concentrations via ventilation, reducing the risk for unsuccessful attempts at standing. Factors that have been implicated in prolonged recovery time include intraoperative hypotension, hypothermia, and duration of anesthesia [19]. A prolonged recovery time is also undesirable, as prolonged recumbency can contribute to the development of myopathies [20], and determining the correlation between trazodone administration and recovery time is important in optimizing patient outcome. This finding supports the use of trazodone in a hospital setting without increasing concern for prolonged recovery following an anesthetic event. The population in this study did not have a standardized anesthesia duration or protocol, which may have impacted this result [21]. Further investigation into recovery time using standardized anesthesia time and protocol is warranted to truly determine if this holds true.

Importantly, there was no significant difference in age or anesthesia duration between groups. This is of importance as these factors have been implicated in affecting recovery outcomes. Longer anesthesia times and older age have been associated with worse recovery quality and increased mortality associated with anesthesia [22,23,24]. Similar results were noted in this population, as longer anesthesia times were found to have a significant effect on recovery time. Anesthesia duration was accounted for as a potential factor by utilizing a population that was undergoing a procedure that, at this institution, results in a similar anesthesia duration. Age was also accounted for as a potential confounding factor that may impact the recovery score by utilizing a case-match control process. Horses were selected as part of the control group based on similar signalment to the case group. By accounting for age and anesthesia time, the impact of known factors that negatively affect recovery may have been minimized.

Complications experienced under anesthesia were noted. Complications noted did not fall outside of the scope of expected anesthetic complications in either group. Hypoxemia and hypotension are common issues experienced during inhalant anesthesia in horses. Previous investigation into the impacts of hypoxemia and hypotension on recovery quality identified a correlation between hypoxemia and worse recoveries, but not hypotension [25]. Furthermore, there was no significant effect of hypotension on recovery times in our population. Hypoxemia was only noted in one horse in this study and therefore is unlikely to have affected the results. Hypocalcemia has been noted to be a common electrolyte abnormality during equine anesthesia. Hypocalcemia may cause muscle weakness and an impact on the recovery period. Investigation of electrolyte disturbances in horses undergoing colic surgery, however, did not find an association between hypocalcemia and increased mortality [26]. Additionally, the horses in this study had appropriate management of these complications, and we believe that this minimized any impact that these factors may have had on the recovery period.

The main limitation of this study is its retrospective nature, allowing for confounding factors to potentially impact the recovery of the horses included in this study. Factors such as anesthetic protocol, individual patient temperament, duration of anesthesia, and individual assessment of recovery scores may have all contributed to the results of this study. In addition, the retrospective nature led to a relatively small sample size. Horses enrolled were identified based on evaluation of medical records without the aim of a total number per group. Based on the small difference in mean recovery score between trazodone and control groups (4.7 ± 0.2 and 4.8 ± 0.1, respectively), post hoc analysis indicates that over 3500 horses would be required to determine statistical significance between groups.

Another limitation is the lack of a standardized anesthetic protocol. Each patient was administered a protocol that was deemed appropriate by an anesthesiologist and this variation in anesthetic protocol may have had an impact on the recovery period [21]. Although anesthetic protocols varied by patient, a majority of horses in both groups were anesthetized with similar protocols and doses. The doses administered to horses in this study are similar to previous work evaluating the effects of trazodone on the anesthetic period [10]. The retrospective nature of this study allows this confounding factor to exist, but utilizing a standardized anesthetic protocol would allow for a level of control over this factor and further prospective studies are warranted.

Additionally, various sedative agents were used during the recovery period. While no significant difference was elucidated in the overall utilization of sedative agents between groups, the agent used, and the timing of administration may have impacted the recovery of these horses. This confounding factor may have impacted the ability of this study to find a difference in recovery score. While the sedative properties of various alpha-2 agonist agents are not equivalent [27], previous studies examining the effects of various alpha-2 agonists in the recovery period did not find a significant difference in recovery when xylazine or dexmedetomidine were used during recovery [28]. Our results may demonstrate that sedative agent administration could play a larger role in the recovery period than trazodone administration prior to anesthesia. This could suggest that trazodone administration prior to anesthesia has less impact than other factors of the anesthetic period. Elucidating the effects of trazodone in the absence of sedative agents in the recovery period is difficult, as removal of these agents entirely would likely have led to increased mortality in the study population [11].

Finally, we utilized a non-validated recovery score. The scale used in this study is a subjective 5-point scale with limited input on the number of variables that are assessed in the recovery period (time to first movement, nystagmus), but it has similarities to recovery grading scales used more commonly in the literature [29]. This scale allows for user bias to have impacted recovery rating, but previous investigations into recovery grading scales have demonstrated that even validated recovery scoring systems resulted in poor agreement between different evaluators [30]. Therefore, although a non-validated scoring system was used, a majority of patients anesthetized for an MRI at this institution are managed by a single dedicated anesthesia technician. As such, we believe that there is limited inter-user variability associated with recovery scores for this patient group, which may add to the accuracy of the scores.

## 5. Conclusions

The results of this study support our hypothesis that trazodone administration prior to an anesthetic event does not appear to impact recovery scores in horses undergoing anesthesia for orthopedic MRI. Furthermore, trazodone administration in this population did not significantly impact recovery duration, administration of sedatives in the recovery period, or administration of additional reversal agents. Administration of trazodone prior to orthopedic MRI did not have an association with poorer recovery in this study, but given the limitations of this study, further prospective investigation is needed to better evaluate the effects of trazodone in the recovery period and determine if this holds true.

## Figures and Tables

**Figure 1 animals-15-02907-f001:**
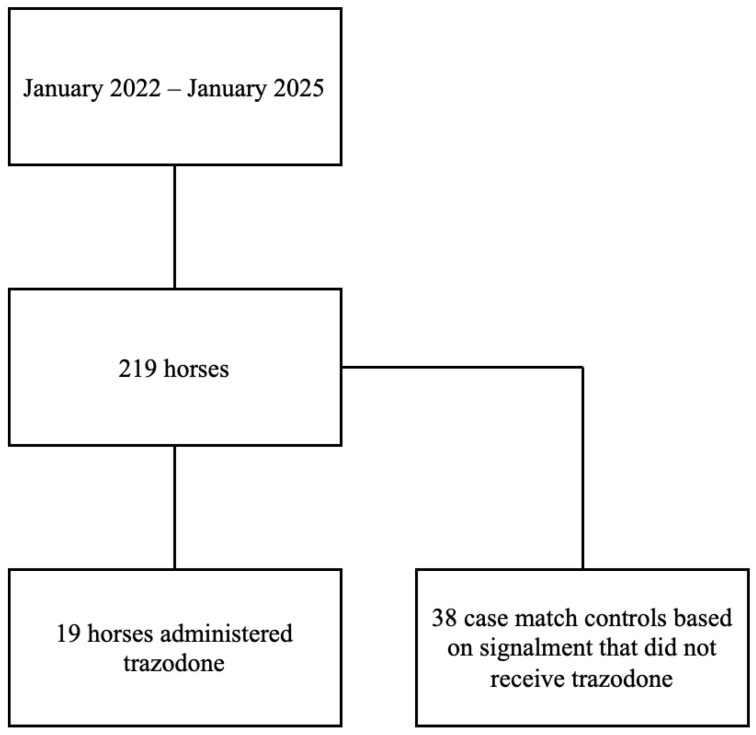
Flow diagram for identification of horses that underwent anesthesia for an orthopedic MRI from January 2022 to January 2025. In total, 19 horses were identified to have received trazodone prior to anesthesia and 38 case-matched control horses were selected.

**Figure 2 animals-15-02907-f002:**
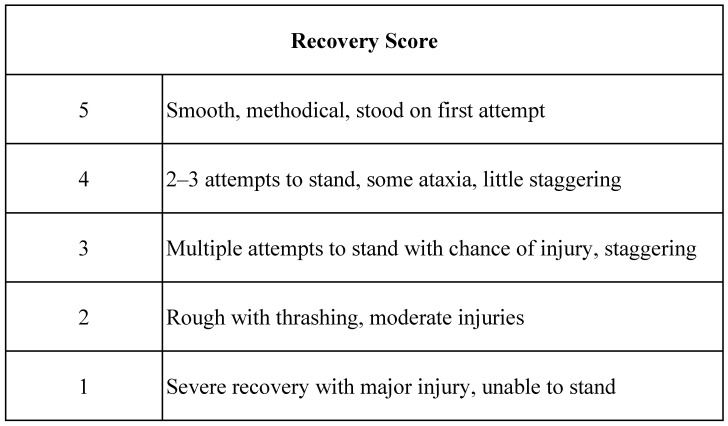
Recovery scoring system utilized.

**Table 1 animals-15-02907-t001:** Signalment of horses that were anesthetized for orthopedic MRI and either did or did not receive trazodone. Age and weight are represented as mean ± SD. There was no significant difference between the groups’ age (*p* = 0.857, 95% CI [−2.54, 2.12]), sex (*p* = 0.195), breed (*p* = 0.265), or weight (*p* = 0.968, 95% CI [−46.62, 48.52]).

	No Trazodone (n = 38)	Trazodone (n = 19)
**Sex**		
F	12	6
M	0	2
MC	26	11
**Breed**		
Andalusian	1	0
Dutch Warm Blood	2	1
German Riding Pony	1	1
Haflinger	1	0
Hanoverian	1	1
Irish Hunter Horse	2	1
Oldenburg	3	1
Paint Horse	1	1
Quarter Horse	8	4
Saddlebred	1	2
Thoroughbred	2	3
Warm Blood	14	3
Welsh Pony	1	0
Westphalian	0	1
**Age**	8.8 ± 3.9	9 ± 4.6
**Weight**	561.9 ± 74.5	560.9 ± 101.9

**Table 2 animals-15-02907-t002:** Recovery scores for horses that either received or did not receive trazodone prior to anesthesia for orthopedic MRI. There was no significant difference between groups (*p* = 0.226, OR = 0.424, CI [0.102, 1.743]).

	Trazodone	
Recovery Score	Y (n = 19)	N (n = 38)
	n	n
3	0	1
4	5	4
5	14	33

## Data Availability

The data that support the findings of this study are openly available in Dryad http://doi.org/10.5061/dryad.4qrfj6qpv, reference number 4qrfj6qpv.

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
