# Peer review of "Preanesthetic Administration of Trazodone Does Not Impact Anesthetic Recovery Scores in Horses"

_animals, 2025, doi:10.3390/ani15192907_

Round 1
Reviewer 1 Report
Comments and Suggestions for Authors
Thank you for preparing this generally well-written manuscript on an interesting topic. I have a few questions for you to consider/which popped into my head as I was reading through:
Methods: Why 19 Trazodone horses and 38 controls if the power calculation suggests fewer cases are sufficient (12 and 24). Did you already suspect your effect size could be smaller than one recovery score?
Results. Table 1. I am not sure you need to repeat the statistical “no difference” results you have already got in the text in the Table legend.
Results. I cannot follow the paragraph regarding post-operative sedation. This states all horses received a sedative and then goes on to say 5/19 horses received a single round of sedation, etc. I think what you mean is all trazodone horses received sedation: 5/19 with a single agent (is this then the xylazine or medetomidine you refer to in the previous line?), 13/19 with two agents and 1/19 with three different agents. Is this correct? Ketamine is an interesting “sedative” in this context, but that is simply a side note on my part.
Results: why did some horses receive atipamezole? Slow recovery/adverse effects of alpha-2 agonist?
Discussion/Limitations: all horses are sedated, and with different agents, meaning the recovery impact of trazodone, which is the primary objective, could be hard to elucidate. Especially if the sedatives employed in recovery are expected to be more sedative in nature than trazodone (consider how long into trazodone administration these recoveries were), it is possible that any effect difference between trazodone/no trazodone is diminished or lost by the routine use of these, often multiple, sedatives. I think this aspect needs further discussion
In general, in the discussion, I feel the findings could be reported with less certainty than the way they are currently presented, especially in light of the many limitations the study- for me this is the unvalidated/unpublished recovery score and the differences in premedications and recovery sedation, and the fact all animals were sedated at least once.
The above limitations clearly arise from a study which is conducted in a clinically relevant circumstance
Author Response
Thank you for taking the time to review our manuscript and we appreciate your input.
Comment 1: Methods: Why 19 Trazodone horses and 38 controls if the power calculation suggests fewer cases are sufficient (12 and 24). Did you already suspect your effect size could be smaller than one recovery score?
Response 1: A prospective power calculation was performed and suggested that 12 and 24 horses would be sufficient. The retrospective nature of the study identified 19 horses during the stated time period who had received trazodone and met the other inclusion criteria. This is the reason we enrolled this number of cases in the study.
Comment 2: Results. Table 1. I am not sure you need to repeat the statistical “no difference” results you have already got in the text in the Table legend.
Response 2: Thank you. This statement has been removed from the manuscript.
Comment 3: Results. I cannot follow the paragraph regarding post-operative sedation. This states all horses received a sedative and then goes on to say 5/19 horses received a single round of sedation, etc. I think what you mean is all trazodone horses received sedation: 5/19 with a single agent (is this then the xylazine or medetomidine you refer to in the previous line?), 13/19 with two agents and 1/19 with three different agents. Is this correct? Ketamine is an interesting “sedative” in this context, but that is simply a side note on my part.
Response 3: Thank you for the clarity of phrasing. We have edited these sentences. We have edited the paragraph to state:
“All horses in both groups received a sedative in the recovery time period. Initial sedative agent used was either dexmedetomidine (n=51) or xylazine (n=6). In 5/19 of horses receiving trazodone only a single sedative agent was administered during recovery (dexmedetomidine in 4/19 [1 mcg/kg, 0.8 - 1 mcg/kg], and xylazine in 1/19 [(0.25 mg/kg]), compared to 17/38 of horses in the control group (dexmedetomidine in 14/38 [1 mcg/kg, 0.8 - 1 mcg/kg)] xylazine in 3/38 [0.36 mg/kg, 0.25 - 0.49 mg/kg]). A second sedative agent was administered to 13/19 horses that received trazodone, compared to 17/38 horses in the control group. Additional sedative agents included xylazine (13/19 horses administered trazodone [0.35 mg/kg, 0.16 - 0.6 mg/kg], 13/38 control horses [0.26 mg/kg, 0.1 - 0.5 mg/kg]), detomidine (single control horse, 0.003 mg/kg), acepromazine (one control horse, 0.018 mg/kg ), and ketamine (one control horse, 0.16 mg/kg). 4/38 horses in the control group received a third sedative agent, compared to 1/19 in the trazodone group. All horses that were administered a third sedative agent were given acepromazine (trazodone group: 0.0015 mg/kg; control group 0.015 mg/kg [0.009 - 0.024 mg/kg]). There was no significant difference (p = 0.577, 95% CI [0.618, 2.250]) in the administration of additional sedation between the two groups. ”
We also agree that ketamine administration was interesting, based on the timing of the administration we suspect that it was likely due to the horse moving during positioning for recovery. We have added this to the discussion, and can be found on page 7.
Comment 4: Results: why did some horses receive atipamezole? Slow recovery/adverse effects of alpha-2 agonist?
Response 4: At the discretion of the anesthesia team, horses that were slow to recover, or were felt to not have made a sufficient attempt at standing in an appropriate time period were administered atipamezole. Atipamezole may help expedite recovery and reduce potential sedative effects [15, Mascarenhas] of alpha-2 agonists administered that may be preventing horses from attempting to stand. This has been added as a discussion point and can be found on page 7.
Comment 5: Discussion/Limitations: all horses are sedated, and with different agents, meaning the recovery impact of trazodone, which is the primary objective, could be hard to elucidate. Especially if the sedatives employed in recovery are expected to be more sedative in nature than trazodone (consider how long into trazodone administration these recoveries were), it is possible that any effect difference between trazodone/no trazodone is diminished or lost by the routine use of these, often multiple, sedatives. I think this aspect needs further discussion.
Response 5: We agree and believe that this may add more strength to our findings. The lack of difference between groups and the impact of the sedative agents on the recovery period may suggest that trazodone administration by itself may not have large or significant impacts on recovery. We have added this to the discussion and can be found on page 8 and 9.
Comment 6: In general, in the discussion, I feel the findings could be reported with less certainty than the way they are currently presented, especially in light of the many limitations the study- for me this is the unvalidated/unpublished recovery score and the differences in premedications and recovery sedation, and the fact all animals were sedated at least once.
The above limitations clearly arise from a study which is conducted in a clinically relevant circumstance
Response 6: Thank you, we agree that the limitations of this clinical retrospective study provides less certainty in the results and we have adjusted phrasing in the discussion and conclusion to reflect this. Thank you again for your efforts.
Reviewer 2 Report
Comments and Suggestions for Authors
Dear Author.
The purpose of this study was to determine whether pre-anesthesia administration of trazodone affects recovery scores in healthy horses undergoing orthopedic magnetic resonance imaging (MRI). Secondary objectives included determining whether there were differences in recovery time, the use of additional sedatives, or the use of antagonists during recovery between horses administered trazodone and those not administered trazodone.
Trazodone is administered to minimize patient anxiety and to correct undesirable behaviors associated with hospitalization and stall rest; however, it can cause ataxia and decreased muscle coordination, which can be problematic during recovery. Previous studies have examined the cardiovascular effects of pre-anesthesia administration of trazodone, but no studies have yet been published that examine the effects of trazodone administration on horse recovery. This study, which focuses on this point, is considered valuable.
However, as the authors note, this study did not consider the influence of other confounding factors, and the number of animals was not considered sufficient for analysis. Therefore, I consider it inappropriate for original research at this time.
Major revision
The dose of trazodone used in this study is not specified. Considering the impact of dosage is important when considering the impact on recovery from anesthesia.
The anesthetic protocols used in the trazodone and control groups were not compared, so their impact on recovery from anesthesia is unknown.
The doses of xylazine, acepromazine, dexmedetomidine, butorphanol, ketamine, propofol, guaifenesin, isoflurane concentration, and dexmedetomidine are unknown.
It is unclear whether these doses are higher or lower than those used in previous studies.
Variables that may lead to shorter standing time include anesthesia duration, hypothermia, and hypotension. The lack of a comparative analysis of these confounding factors makes this article difficult to write. It is also necessary to provide more detailed case data.
Minor revision
Have you examined the usefulness of the scoring system in Figure 2? You mentioned that it is similar to a more commonly used recovery assessment scale. Please explain if you have compared the two or used them as references when creating your scoring system.
The use of trazodone in horses can lead to ataxia and decreased muscle coordination [8], raising concerns about the risk of complications during recovery from anesthesia. This study suggests that the use of trazodone in hospitals may not adversely affect recovery time, but are there any differences from Study 8, such as dosage or administration method? What do you think accounted for the lack of adverse effects?
Author Response
Comment 1: The purpose of this study was to determine whether pre-anesthesia administration of trazodone affects recovery scores in healthy horses undergoing orthopedic magnetic resonance imaging (MRI). Secondary objectives included determining whether there were differences in recovery time, the use of additional sedatives, or the use of antagonists during recovery between horses administered trazodone and those not administered trazodone.
Trazodone is administered to minimize patient anxiety and to correct undesirable behaviors associated with hospitalization and stall rest; however, it can cause ataxia and decreased muscle coordination, which can be problematic during recovery. Previous studies have examined the cardiovascular effects of pre-anesthesia administration of trazodone, but no studies have yet been published that examine the effects of trazodone administration on horse recovery. This study, which focuses on this point, is considered valuable.
However, as the authors note, this study did not consider the influence of other confounding factors, and the number of animals was not considered sufficient for analysis. Therefore, I consider it inappropriate for original research at this time.
Response 1: Thank you for taking the time to read and evaluate our manuscript, we appreciate your feedback.
Based on a prospective power analysis, it was determined that 12 horses administered trazodone and 24 control horses would be sufficient for this retrospective study, therefore we enrolled 19 horses in the trazodone group and 38 in the control group. A post-hoc analysis after data collection then identified that over 3,500 horses would be required to have identified a significant difference in recovery score. We recognize that there are a number of confounding factors that could have affected the results of this study, but these factors cannot be controlled for entirely in a retrospective study. The information provided from this study provides foundational data that can be expanded on and allow for formation of prospectively designed studies that could control these confounding factors.
Comment 2: The dose of trazodone used in this study is not specified. Considering the impact of dosage is important when considering the impact on recovery from anesthesia.
Response 2: Thank you for this insight. We have added the mean dose administered and can be found in the results section on page 4.
Comment 3: The anesthetic protocols used in the trazodone and control groups were not compared, so their impact on recovery from anesthesia is unknown.
Response 3: We utilize a relatively standard anesthetic protocol that has minor variations amongst horses mainly in regards to dose and timing of alpha-2 agonists. Therefore, although the anesthetic protocols were not truly controlled for, a majority of these horses received similar protocols. As this is a retrospective study, we attempted to utilize a population of horses where anesthetic protocol impact may be minimized. We have added discussion points on page 8 and 9.
Comment 4: The doses of xylazine, acepromazine, dexmedetomidine, butorphanol, ketamine, propofol, guaifenesin, isoflurane concentration, and dexmedetomidine are unknown.
Response 4: We have added these to the manuscript. For the dexmedetomidine infusion, the total amount administered was not available for all horses, the data that was recorded (rate of infusion) on the medical records was reported. This can be found in the results section on page 5 and 6.
Comment 5: It is unclear whether these doses are higher or lower than those used in previous studies.
Response 5: Thank you for this suggestion. The mean doses used in this study are considered typical for equine anesthesia, and is similar to doses of anesthetic agents used in previous studies examining the relationship between trazodone administration and anesthesia. This information has been added to the discussion. This can be found on page 9.
Comment 6: Variables that may lead to shorter standing time include anesthesia duration, hypothermia, and hypotension. The lack of a comparative analysis of these confounding factors makes this article difficult to write. It is also necessary to provide more detailed case data.
Response 6: Anesthesia duration has been compared in this study and there was no significant difference between groups, therefore we feel that this factor has been adequately controlled for. Temperature is not routinely monitored and heat support is usually not provided in adult equine patients at this institution. Hypothermia was likely experienced by both groups, and therefore we suspect affected both groups equally. Additionally, hypotension experienced under anesthesia was short in duration and managed in a timely manner. We have added analysis comparing effects of hypotension on recovery time (this can be found in the methods section on page 4), and found that hypotension was not significantly associated with longer recovery times (this can be found in the results section on page 6). Hypoxemia was only noted in one patient in this population, and is unlikely to have affected the results. With these finding, we believe that this also is unlikely to have affected recovery of one group over the other.
Comment 7: Have you examined the usefulness of the scoring system in Figure 2? You mentioned that it is similar to a more commonly used recovery assessment scale. Please explain if you have compared the two or used them as references when creating your scoring system.
Response 7: This is the traditional standard recovery score used at our institution. We recognize the shortcomings of using an unpublished recovery score but this was the data available to us in this retrospective data set. We reference a previously published scoring system that has strong similarities to the recovery scoring system used at this institution, but did not use this when creating ours.
Comment 8: The use of trazodone in horses can lead to ataxia and decreased muscle coordination [8], raising concerns about the risk of complications during recovery from anesthesia. This study suggests that the use of trazodone in hospitals may not adversely affect recovery time, but are there any differences from Study 8, such as dosage or administration method? What do you think accounted for the lack of adverse effects?
Response 8: The study performed by Davis et al. found that mild to moderate sedation and mild ataxia occurred at the oral doses used (7.5 and 10 mg/kg). Sedation and ataxia was noted to have a duration of 4-8 hours. The doses used in the previous study are higher than the doses used in our population. Based on pharmacokinetic and pharmacodynamic data collected by Kynch et al [17], where oral dose of trazodone (4 mg/kg) was closer to our population (3.3 mg/kg), we suspect our population experienced similar effects to the horses in that study. Horses administered 4 mg/kg of oral trazodone experienced mild sedation with a duration of up to 6 hours. We suspect that there was a lack of adverse effects on the recovery period due to the number of other factors that are known to have significant impacts on recovery. We suspect that the lack of difference found in this study reflects this.
Round 2
Reviewer 2 Report
Comments and Suggestions for Authors
This manuscript has been appropriately revised in response to the reviewers' comments, and therefore is deemed acceptable for publication in Animals.